# Shades of Fear—Mental and Physical Health Reactions to the COVID-19 Pandemic: A Representative Study of Polish Society

**DOI:** 10.3390/ijerph20032245

**Published:** 2023-01-27

**Authors:** Tomasz Sobierajski, Marek Krzystanek

**Affiliations:** 1Faculty of Applied Social Sciences and Resocialization, Warsaw University, 26/28 Krakowskie Przedmieście Str., 00-927 Warsaw, Poland; 2Clinic of Psychiatric Rehabilitation, Department of Psychiatry and Psychotherapy, Faculty of Medical Sciences in Katowice, Medical University of Silesia, 45/47 Ziołowa Str., 40-635 Katowice, Poland

**Keywords:** despondency, insomnia, older adults, public health, SARS-CoV-2, young adults

## Abstract

The study was carried out one year after the establishment of the pandemic state in the European Union (EU), the situation at the end of the next wave of the COVID-19 pandemic in Poland. The survey was conducted on a representative sample of Polish people using Computer Assisted Web Interviewing (CAWI), considering several demographic categories, such as sex, age, place of residence, education, and monthly income. The survey’s main objective was to find out whether the respondents feel fear related to pandemics and living in a pandemic, and if so, what the psychological and physiological symptoms of this fear are. Half of the respondents (50.2%) declared that they felt fear about what their life would look like after the pandemic, and every tenth person (10.1%) marked the highest level of fear on the scale. The respondents felt the psychological symptoms of the pandemic much more often than they felt the physical ones. The most common psychological symptoms were fear of the future (38.5% of them gave “often” and “very often” responses, together), despondency (29.2% of them gave “often” and “very often” responses), and mental tension (28.9% of them gave “often” and “very often” responses). A detailed analysis of data from representative studies showed that the responses in a pandemic are strongly determined by demographic categories, mainly sex and age, and they differ depending on the social group to which a person belongs.

## 1. Introduction

The COVID-19 pandemic is a social stressor that has acted on a global scale. A significant number of illnesses and deaths around the world, lockdowns, and isolation, followed by social distancing and limitations to movement and access to medical aid, the paralysis of health care, job losses, and an economic crisis caused individual and social adaptive and maladaptive responses in the form of emerging mental health problems or exacerbation of mental disorders symptoms have occurred [1,2,3]. A meta-analysis by Salari et al. indicated that the COVID-19 pandemic is affecting people’s mental health, albeit to varying degrees, for different social groups [4].

The long-term effects of this global social distress are difficult to predict. However, during a pandemic, it is essential to understand the behavior of society, and within it, individual social groups that that have differences due to demographics, and plan a strategy to deal with similar situations in the future [5]. The experience of other pandemics shows that the impact of a pandemic on people’s mental health is tremendous, and its psychosocial consequences are too complicated to predict [6,7,8]. The COVID-19 pandemic is a political, economic, and health crisis and a social experiment in its own right [9], the analysis of which requires collaborative, interdisciplinary effort between psychiatry and the social sciences.

People’s psychosociological and psychophysical reactions in the first weeks of this pandemic crisis have been studied [10,11,12,13]. However, there are still not so many studies on the social consequences of the pandemic in its subsequent stages, i.e., one year after the announcement of the global lockdown, and the results that have already been published are preliminary; this is especially true for the limited number of population studies [14,15].

For this reason, the main goal of the study was to find out whether one year after the announcement of the global COVID-19 pandemic (March 11, 2020), people experienced fear related to the pandemic and life in the pandemic, and if so, what the psychological and physiological symptoms of this fear were and how the individual symptoms correlated with demographic groups in the population.

In May 2020, during the first stage of the COVID-19 pandemic, a study was conducted on a representative sample of Polish residents, which reported that mental health deterioration due to the pandemic was independently correlated with the female sex and higher education [10]. A British study by Pierce et al. conducted a few weeks after the beginning of the pandemic in the UK was a longitudinal household cohort study, which indicated that clinically significant levels of mental distress increased across the population, with the increases being most significant among the youngest 18–24-year-old people and women [11]. With these studies in mind, and considering the long unprecedented scale of social distress in the form of the COVID-19 pandemic and its potentially life-threatening nature, the authors decided to find answers to several research questions:Do people feel the fear associated with a pandemic and life after a pandemic?How does the COVID-19 pandemic affect people’s well-being?Which fear symptoms, psychological and physiological ones, are most commonly experienced during the pandemic and after the pandemic?How does the pandemic trigger fear in people in different social groups?How do young adults respond to the stressors of a pandemic?

Based on the available knowledge and scientific and research experience, the authors decided to make two research hypotheses:

**H1.** 
*The COVID-19 pandemic had a greater negative psychological impact on the youngest group of adults in society.*


**H2.** 
*People one year after the pandemic outbreak will be more likely to experience psychological than physiological symptoms of fear.*


## 2. First Year of the COVID-19 Pandemic in Poland

### 2.1. Epidemiology of COVID-19

The COVID-19 pandemic influenced the population in Poland in 2021. In the first half of 2021, over 270,000 people died. Compared to the first half of 2020, the number of deaths increased by approximately 62,000, and the death rate amounted to 14.2%. March and April in 2021 were particularly critical periods in terms of the number of deaths in Poland, as there were nearly 14,000 deaths a week, which should be directly related to the impact of the COVID-19 pandemic. The decline in the population in Poland is also the result of the natural decline that has been observed in Poland since 2013, resulting from the low number of births, with a simultaneous increase in deaths resulting from the increase in the number and percentage of older adults. As a result, the total population in Poland decreased by 103,000 people in the first half of 2021.

Women constitute 53% of the total Polish population. The feminization rate is 107 (111 in cities; 101 in the countryside). The average age for males is 40.1 years, and for females, it is 43.3 years. Every fifth person (20.1%) is under 20 years of age, and 18.6% of people are 65 or older. The pre-working age is 18.2%, the working age is 59.5%, and the post-working age is 22.3%. According to the aggregated data shown in the so-called “age pyramid”, which in the case of Poland is regressive, Polish society is an aging society. Most Polish people (59.9%) live in cities, of which nearly every fourth person (23.6%) lives in a city with more than 200,000 inhabitants. The average monthly salary in Poland at the time of the study was PLN 4249 per person. Higher than average earnings were received by 31% of women and 37.5% of men. The minimum income was PLN 2061 per person (approximately $ 542 or EUR 448).

### 2.2. Calendar of Restrictions in the First Year of the COVID-19 Pandemic

The epidemic in Poland was officially declared on March 20. However, all types of schools and all cultural institutions were closed as early as March 12. Very restrictive restrictions were introduced on March 24, limiting the size of gatherings to two people except for family, work, and religious places, and limiting the ability to go outdoors. In the following days, boulevards, beaches, and forests were closed entirely. Religious gatherings were limited to five people, and airports were closed. In mid-April, an order was introduced for people to cover their nose and mouth in public places. In early May, restrictions began to be lifted one by one (mainly due to the possibility of a presidential campaign and general elections), and the country was divided into three zones: red ones with the strictest restrictions, yellow ones with medium-level restrictions, and green ones with low-level restrictions. The zone designation was related to the number of COVID-19 infection cases in a given county. At the end of October, due to the increasing number of cases of COVID-19 infections, the entire country was designated as a red zone, and schools, among other institutions, were again closed. As of December 28, the stabilization stage, i.e., the division of Poland into three zones (red, yellow, and green ones) was returned. From the beginning of 2021, the restrictions were gradually lifted in connection with introducing a universal vaccination against COVID-19.

## 3. Methods

### 3.1. Design of the Study and Sample Size

The study was a social, sociological research embedded in a public health survey framework. The study was designed based on the methodological assumptions of the quantitative method in social research. For this reason, psychological scales were not used to determine the scale of specific symptoms. The survey was carried out by one of the leading Polish research agencies, SW Research, on March 2021 on a representative sample of 1.000 Polish people under a random selection condition to represent the nature of the population accurately using CAWI (Computer Assisted Web Interviewing). The survey questionnaire was delivered to the respondents via a link on the “SW Research Panel”, SW Research’s online research platform. The sample was stratified in layers. In this study, the layers were: sex, age, education, place of residence, and income. Object representativeness assumes that, apart from the layers mentioned above, no other significant variables could influence the results obtained in the study. A detailed description of the study population is provided in Table 1.

### 3.2. Questionnaire Design

The questionnaire used in this study was originally designed for this study, following the latest sociological and methodological knowledge of the authors. The questionnaire consisted of 20 questions: 5 metric questions and 15 factual questions. The metric questions asked about sex, age, education, place of residence, and income. The pertinent questions asked about the fear associated with life during and after a pandemic and the psychological and physiological symptoms of the fear. Based on the available literature, including that relating to predictors of stress in the face of past pandemics [7,16,17,18] and the experience and knowledge of the study’s authors, it was arbitrarily decided to examine nine symptoms of fear: despondency, mental tension, concerns about the future, insomnia, concentration problems, depressing moods, breathing problems, chest pain, and digestive problems. Each question was closed. Answers to some of the questions were presented on a scale. For the metric questions, the scale was adjusted according to the scope of the question, such as the size of the locality or educational level. For the factual questions, the respondents were asked to specify how rarely/frequently they felt a particular symptom due to the pandemic. For the factual questions, an extensive Likert scale was used. For the symptom questions, a 6-degree scale was used, where one end of the scale meant “very rarely” and the other meant “very often.” In addition, for the questions about symptoms, the respondents could respond with the answer “I do not feel it at all.” Other questions used a 7-point scale, where one end of the scale meant “strongly disagree” and the other end of the scale meant “strongly agree”, with an intermediate answer of “neither agree nor disagree.”

The questionnaire was validated ad hoc, revised by qualified interdisciplinary experts, and then adapted to the technique used in the study. The questionnaire was extensively evaluated for the implementation of the study. For this purpose, a pilot study on a random group of 20 respondents was conducted to verify the correctness of the tool. The study group from the pilot study was not included in the main study group. After considering the methodological and technical comments, the questionnaire was technically adapted and given to the research panel of the research company, and then sent to the randomly selected respondents via a link.

### 3.3. Statistical Analysis

Statistical analyzes were performed in IBM SPSS Statistics 27.0.1.0. Data for all of the outcomes are reported for all of the participants. The relationship between the variables was evaluated by using the Chi-squared test. The Kruskal–Wallis’ test was used to analyze the questions using the Likert scale. Answers to the questions are presented with the total number of respondents (n) and frequencies of the subgroups (%). For all of the analyzes, a *p*-level of <0.05 was considered to be statistically significant.

### 3.4. Ethical Considerations

The company that carried out the research is a member of ESOMAR (European Society for Opinion and Marketing Research), and it provides the guarantee of the ethical implementation of the research and the protection of the respondents’ data. The quality of the tests and compliance of the test procedures with the standards was confirmed by the PKJPA (quality control program for interviewers) quality certificate granted to SW Research in 2015. All of the survey participants gave informed consent to participate in the study. The study was conducted following all of the ethical rules in Poland and the European Union relating to the implementation of social research. No individual-level data were used, and no data can be linked to any individual.

## 4. Results

### 4.1. Participants Characteristics

The study was conducted on a representative sample of 1000 adult Polish people, most of whom were women (N = 531, 53.1%), and the most common age group were people between 50 and 64 years of age (N = 309. 30.9%); nearly half of the respondents had secondary education (N = 460, 46.0%), 6 out of 10 people lived in the city (N = 599, 59.9%), and the most common groups of people earned between PLN 2000–3000 per month (N = 231, 23.1%). The detailed demographic distribution of the studied group, corresponding to the quota distribution of demographic variables in individual categories, is presented in Table 1.

### 4.2. Fear of Life after the Pandemic

Half of the respondents (N = 502, 50.2%) declared that they felt fear about what their life would be like after the pandemic, with every tenth person (N = 101, 10.1%) noting they felt the highest level of fear on the scale. Every ninth respondent (N = 118, 11.8%) declared that they had no fear of what their life would look like after the pandemic. Every fourth person (N = 250, 25.0%) had ambivalent feelings about their future after the pandemic, and it was unable to determine whether they feel fear.

Regarding the sense of fear, there were no significant differences in the groups (Table 2 and Table 3).

### 4.3. Psychological and Physiological Symptoms of Fear

Respondents were asked how often (and if at all) they experienced the following symptoms due to the pandemic: despondency, mental tension, fear for the future, insomnia, difficulty concentrating, depressive moods, breathing problems, chest pain, and digestive problems. The most common symptom was fear of the future (38.5% of them gave “often” and “very often” responses), which did not occur only in a small group of respondents (4.5%). The next most frequently indicated symptoms were despondency (29.2% of them gave “often” and “very often” responses), which was not felt at all in 8% of the respondents, and mental tension (28.9% of them gave “often” and “very often” responses), which was not generally felt by every tenth respondent (10.3%). The least frequent symptoms were breathing problems, which did not occur at all in every third respondent (32.2%), chest pains (28.4% of them answered “I did not feel at all”), and digestive problems, which every fifth person did not experience (20.6%) (Figure 1).

#### 4.3.1. Despondency

Due to the pandemic, despondency was much more frequently experienced by women than it was men (15% vs. 9% of them gave “very often” responses) (*p* < 0.001) and young people up to 24 years of age (18.5% of them answered “very often”) (*p* = 0.014) (Table 4).

#### 4.3.2. Mental Tension

Due to the pandemic, mental tension was felt more often by women than it was men (14.1% vs. 9.6% of them gave “very often” responses) (*p* = 0.011) and respondents up to the age of 50 years old than it was by those over 50 years old (16.2% of people up to 24 years old, 15% of people between 25 and 34 years old, 14.3% of people between 35 and 49 years old vs. 8.1% of people between 50 and 64 years old and 8.5% of people over 65 who gave “very often” responses (*p* = 0.002) (Table 5).

#### 4.3.3. Fears for the Future

Due to the pandemic, fears for the future were felt much more often by women than they were felt by men (22.1% vs. 14.5% of them gave “very often” responses) (*p* < 0.001), and people with the lowest income up to PLN 1000 (31.4% of them gave “very often” responses) (*p* = 0.008) (Table 6).

#### 4.3.4. Insomnia

Due to the pandemic, insomnia was experienced slightly more often by women than it was by men (11% vs. 7.7% of them gave “very often” responses) (*p* = 0.030) (Table 7).

#### 4.3.5. Difficulty Concentrating

Due to the pandemic, a problem with concentration was experienced more often by respondents up to 49 years old than it was by respondents aged 50 years old and older (respectively: 11.8% people up to 24 years old, 9.5 people aged between 25 and 34, 9.9% people aged between 35 and 49, 4.8% people aged between 50 and 64, and 5.3% people aged 65 and over gave “very often” responses (*p* < 0.001) (Table 8).

#### 4.3.6. Depressive Mood

Due to the pandemic, a depressive mood was slightly more frequently experienced by women than it was by men (*p* = 0.050), and it was significantly more often experienced by young people up to 24 years old (16.4% of them gave “very often” responses) (*p* = 0.010) (Table 9).

#### 4.3.7. Respiratory Problems

There were no statistically significant correlations between the demographic groups in terms of respiratory problems due to the pandemic (Table 10).

#### 4.3.8. Chest Pain

Due to the pandemic, chest pain was very commonly experienced by respondents with primary education (*p* = 0.048) (Table 11).

#### 4.3.9. Digestive Problems

There were no statistically significant correlations between the demographic groups in terms of digestive problems due to the pandemic (Table 12).

## 5. Discussion

Our nationally representative survey showed that the COVID-19 pandemic impacted Polish people’ mental state, albeit with varying degrees of severity, in different social groups. The scientists who studied and analyzed the mental health of societies affected by the SARS and Ebola viruses, which had a very dynamic and severe course in Asia and Africa at the beginning of the 21st century, reached similar conclusions [19,20,21,22,23]. The studies that refer to the mental state of people in the first stage of the SARS-CoV-2 pandemic, when the global lockdown was enforced, also indicated that there was a threat to health and life caused by social distancing, isolation, and elevated stress, which led to increased anxiety and depressive symptoms [24,25,26,27,28,29].

Our study showed that the youngest respondents under 24 years of age were the most likely out of all of the age groups to experience fear symptoms such as despondence (*p* < 0.001), mental tension (*p* = 0.011), concentration problems (*p* < 0.001), and depressive moods (*p* = 0.010), thereby confirming H1. The COVID-19 pandemic had a more significant negative psychological impact on the youngest adult population. A report by the British Parliamentary Group on a Fit and Healthy Childhood referred to the youngest members of society as the “COVID Generation” because a range of data, e.g., remote learning, severance of social ties, the mental state of children before the pandemic, and the predictors created based on that data, indicate that the COVID-19 pandemic will affect the long-term feelings of children, teenagers, and young adults [30]. It was confirmed by the aforementioned British study conducted during the first period of the pandemic [11]. A Japanese study indicated that while the first months of the pandemic resulted in a reduction in the overall suicide rates, suicide rates increased again in the second half of it, with the most significant increases being among women, children, and young adults [31]. In research on the Brazilian population, the youngest generation of respondents also reported a higher level of anxiety than the other age groups did [32]. Additionally, a meta-analysis collecting data from studies of the first year of the COVID-19 pandemic from 204 countries around the world confirms that younger age groups are more prone to anxiety disorders due to the pandemic than older age groups are [33]. The WHO report “*Mental Health and COVID-19*” also found a similar results [34].

On the other hand, older adults over 50 years old, such as in a Serbian study [24], similar to respondents with higher education and those from the most significant cities, complained of depressive moods, despondency, or fear of the future much less frequently. It may indicate a lower resilience of younger people in a pandemic, with there being a lower reactivity to stress among the elderly people [35,36]. Increased fear due to a pandemic may translate into an increased number and frequency of depressive states in young people, ultimately affecting their future lives. It is hypothesized that adolescents and young adults who are depressed are more likely than adolescents and young adults who are not depressed to fail to complete their education and become unemployed [37,38,39].

In our study, mental symptoms were experienced by the respondents more often than physical ones were (Figure 1), which positively verifies H2. People one year after a pandemic outbreak will be more likely to experience mental than physiological symptoms of fear. The relatively lower frequency of somatic anxiety symptoms compared with that of depressive symptoms may be related to the fact that the study was conducted during an ongoing pandemic, which mobilized the ability to adapt to the situation of distress [40,41]. Perhaps the frequency of anxiety disorders may increase after the distress situation is over, but to achieve confirmation of this hypothesis, we require follow-up studies.

The presence of a depressive mood was confirmed by 24.5% of the respondents, with statistically significant differences being observed according to sex (*p* = 0.050), with women experiencing a depressive mood more often than men did (*p* = 0.010). This result is consistent with the results of the meta-analysis based on data from the first stage of the pandemic, which indicated that the estimated frequency of depression occurred on average in 25% of the respondents [42]. A systematic review conducted by Xion et al. from the pandemic outbreak to May 2020 referring to general population surveys in eight countries around the world indicated a marked increase in drug use rates, depressive moods, and even post-traumatic stress disorder [43].

Insomnia occurring very often or often was experienced by 22.9% of respondents in our study, and it statistically affected women more often than it affected men (*p* = 0.030). In Greek studies, 37.6% of the respondents reported that they had insomnia. However, this study was carried out in the first weeks of the pandemic [44]. In a Portuguese study that was also conducted in the first weeks of the global lockdown, 4.5% of the respondents had increasingly severe insomnia, and 22.9% had stable, severe insomnia [45]. It could indicate a possible persistence of sleep disturbances despite the duration of the pandemic. An international, multicenter study by Morin et al. conducted from May to August 2020 indicated that insomnia was more common in women, as well as in younger age groups and the residents of Brazil, Canada, Norway, Poland, the U.S., and the U.K. Interestingly, the number of insomnia cases was higher among those who completed the questionnaire at the beginning of the study than it was for those who completed the questionnaire at the end of the study [46]. Canadian researchers who compared the insomnia rates of respondents in 2018 and 2020 in a panel study found that during the COVID-19 pandemic, there was a significant deterioration in sleep quality and an emergence of other symptoms such as despondence, fear, and depressive moods [47].

In Turkish [48], American [49], Spanish [50], and Australian [51] studies carried out during the pandemic, women experienced more fear of the pandemic than men did. Our study indicated statistically significant relationships between sex and despondence (*p* < 0.001), mental tension (*p* = 0.011), fears for the future (*p* < 0.001), insomnia (*p* = 0.030), and depressive moods (*p* = 0.050). All of these symptoms were experienced more often by women than they were by men. The meta-analysis mentioned above, which collected data from most of the countries and regions in the world, found that females were affected more by the pandemic than males were in terms of depressive and anxiety disorders [33].

The studies conducted around the world indicate that the level of anxiety was strongly correlated with the age of the women surveyed and the sense of responsibility for the other person in the family, as indicated, for example, by a Brazilian study that indicated a strong correlation between fear and the age of the women. The more the women were entered into interpersonal relationships, the greater the fear of infection was. It has also been indicated that there is a higher level of fear among those women who are responsible for their loved ones and live in a family [52]. It is especially true for mothers of young children and pregnant women, as confirmed by studies and meta-analyses conducted in many countries around the world [53,54,55,56,57]. Meanwhile, in a U.S. study of a non-representative sample of middle-aged women (45–55), no significant differences were found in the mental and physical health measures among the women that were surveyed before and during the pandemic. An analysis of open-ended questions found that while some women reported increased stress levels, others emphasized that the pandemic gave them more time for themselves and their families [58].

### Strengths and Limitations of the Study

The present study had several limitations. It was carried out after one year of the pandemic when vaccinations against COVID-19 were already available in the EU, including Poland. At the time of the study in Poland, only the oldest people and selected professional groups, such as teachers and doctors, could be vaccinated. After some of the population was vaccination, the tension and fear of the pandemic may have decreased or increased in some groups. Since this study, such as many others, showed that demographic categories are essential variables of the sense of psychological threat and fear related to a pandemic, in subsequent studies, it is worth extending the demographic analysis to include marital status and having/not having children. The study was only conducted over a short period, so it does not reveal the dynamics of the occurrence of the observed symptoms, which could facilitate the interpretation of the obtained results.

An interesting explanation for the higher levels of anxiety among the young people and lower levels of anxiety among the elderly people could be a thesis related to the consumption of a particular type of media from which respondents obtained knowledge about the pandemic. It can be assumed with a high degree of probability that young people are exposed to new media more often, i.e., social networks, where a lot of disinformation could be found, which could influence the elevated level of anxiety. Older people are more likely to use traditional media (TV and radio), where the information is verified, and much less disinformation appears. However, in our study, we did not examine the sources of the information obtained, so we can only point to this thesis as an area for further in-depth research.

The strengths of this study include the fact that it was carried out on a representative, nationwide group based on the best methodology for conducting social research using an original questionnaire that was explicitly created for this study. This choice was caused by the desire to investigate the society’s specific reaction and adaptation to the pandemic situation based on the sociological and psychological knowledge of the article’s authors. The survey research results of a representative group of Polish society can be extrapolated to demographic groups with great certainty. Consequently, the results of a nationwide, representative study conducted in a large European country such as Poland can be used for estimations in other European countries, especially those with a similar population structure. Based on the obtained data, it is possible to build socio-psychological models of society’s reaction and adaptation to a pandemic situation, allowing for planning health and political strategies to manage the subsequent mass health crises.

## 6. Conclusions

Based on our results, an emerging thesis requires further research and in-depth analysis: the optimistic well-being of young adults does not translate into their mental state during a pandemic. It does not eliminate the high level of fear that the pandemic will have an impact. At the same time, the optimistic self-esteem of the mental state of older adults may translate into a reduction in the level of fear and psychological feelings caused by the pandemic. It also indicates the greater resilience of older adults compared to that of younger adults in a pandemic situation.

The survey results of a representative group of adult Polish people can help to develop survival strategies and create social policies to support people and social groups in future pandemic crises.

## Figures and Tables

**Figure 1 ijerph-20-02245-f001:**
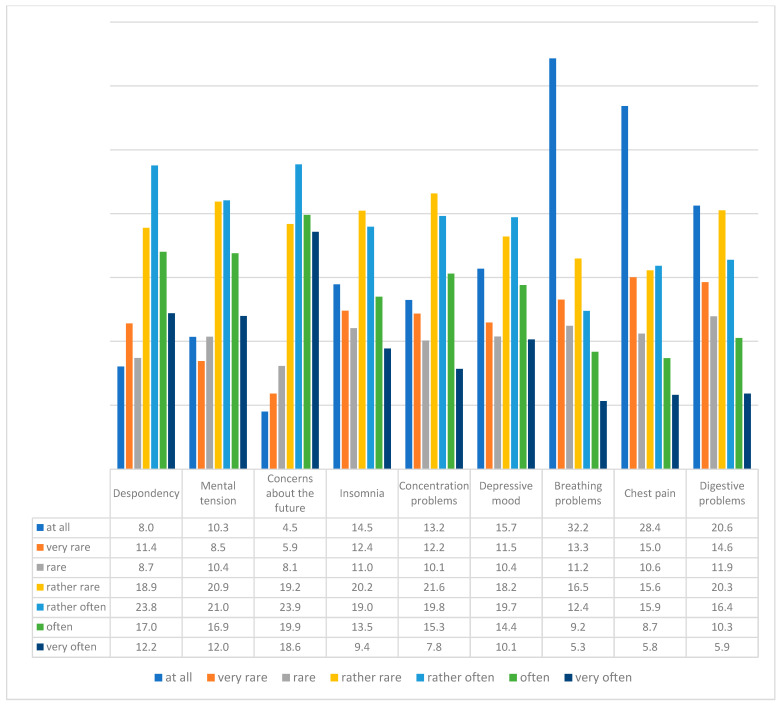
Feeling of psychological and physiological symptoms of fear due to the pandemic (N = 1000).

**Table 1 ijerph-20-02245-t001:** Sociodemographic characteristics of survey participants, N = 1000.

	N (%)
**Total**	1000 (100)
**Sex**	
Female	531 (53.1)
Male	469 (46.9)
**Age**	
18–24	124 (12.4)
25–34	190 (19.0)
35–49	243 (24.3)
50–64	309 (30.9)
65 and more	135 (13.5)
**Education**	
Elementary (primary)	32 (3.2)
Vocational	91 (9.1)
Secondary	460 (46.0)
Tertiary (with university diplomas)	418 (41.8)
**Place of residence**	
Village (rural area)	401 (40.1)
Town to 20 k citizens	101 (10.1)
Town of 20–99 k citizens	205 (20.5)
City of 100–199 k citizens	85 (8.5)
City of 200–499 k citizens	87 (8.7)
City over 500 k citizens	121 (12.1)
**Income (monthly)**	
Up to 1000 PLN	55 (5.5)
1000–2000 PLN	209 (20.9)
2001–3000 PLN	231 (23.1)
3001–5000 PLN	18 (19.8)
Over 5000 PLN	100 (10.0)
Refusal to answer	207 (20.7)

**Table 2 ijerph-20-02245-t002:** Do you feel any fear of what your life will be like after the pandemic? (Summary of responses by demographic categories) (N = 1000).

	Definitely, I Do Not	I Do Not	Rather, I Do Not	I Neither Do Nor Do Not	Rather, I Do	I Do	Definitely, I Do
N (%)							
**Sex**							
Female	58 (11.0)	28 (5.3)	34 (6.4)	132 (24.9)	138 (26.0)	80 (11.3)	60 (11.3)
Male	60 (12.8)	28 (6.0)	40 (8.4)	117 (25.0)	116 (24.7)	68 (14.5)	41 (8.7)
**Age**							
18–24	14 (11.5)	6 (4.6)	12 (9.6)	31 (25.0)	29 (23.1)	22 (17.8)	10 (8.4)
25–34	18 (9.5)	6 (3.0)	13 (6.7)	51 (26.8)	59 (31.3)	17 (9.0)	26 (13.6)
35–49	28 (11.3)	15 (6.1)	22 (9.1)	58 (24.1)	56 (23.0)	33 (13.7)	31 (12.6)
50–64	38 (12.2)	19 (6.1)	21 (6.7)	79 (25.5)	77 (24.9)	55 (17.7)	21 (6.8)
65 and more	21 (15.3)	11 (8.3)	6 (4.6)	30 (22.5)	33 (24.4)	20 (15.1)	13 (9.9)
**Education**							
Primary	3 (9.3)	2 (5.9)	4 (12.9)	11 (35.1)	7 (21.7)	3 (9.2)	2 (6.0)
Vocational	9 (10.0)	2 (2.2)	7 (7.8)	21 (23.5)	25 (27.5)	15 (16.7)	11 (12.3)
Secondary	59 (12.7)	29 (6.4)	30 (6.6)	117 (25.4)	117 (25.4)	56 (12.2)	52 (11.4)
Tertiary	48 (11.4)	23 (5.6)	32 (7.7)	100 (24.0)	105 (25.2)	73 (17.5)	36 (8.6)
**Place of residence**							
Village (rural area)	39 (9.7)	22 (5.5)	33 (8.3)	105 (26.2)	109 (27.2)	52 (12.9)	40 (10.1)
Town to 20 k citizens	9 (8.8)	4 (3.8)	11 (10.7)	20 (20.1)	25 (24.8)	20 (19.5)	12 (12.4)
Town of 20–99 k citizens	28 (13.7)	11 (5.3)	12 (5.6)	51 (24.7)	47 (22.8)	33 (16.0)	24 (11.8)
City of 100–199 k citizens	12 (13.7)	7 (8.0)	6 (7.1)	22 (26.0)	20 (23.2)	13 (15.3)	6 (6.7)
City of 200–499 k citizens	13 (14.5)	2 (2.3)	6 (6.8)	21 (23.7)	23 (26.9)	15 (17.0)	8 (8.7)
City over 500 k citizens	18 (14.7)	11 (8.8)	6 (4.8)	31 (25.4)	30 (24.4)	15 (12.8)	11 (9.1)
**Income (PLN)**							
>1000	5 (9.2)	1 (1.7)	3 (5.6)	16 (28.5)	11 (20.4)	10 (18.5)	9 (16.1)
1000–2000	22 (10.5)	16 (7.7)	16 (7.7)	48 (22.9)	58 (27.9)	27 (12.8)	22 (10.5)
2001–3000	39 (16.9)	13 (5.5)	11 (4.7)	51 (22.3)	61 (26.4)	31 (13.4)	25 (10.6)
3001–5000	21 (10.4)	15 (7.7)	14 (6.9)	52 (26.3)	54 (27.4)	32 (16.2)	10 (5.1)
<5000	10 (9.9)	3 (2.9)	12 (12.1)	31 (30.8)	21 (21.4)	12 (11.9)	11 (11.0)
Refusal to answer	22 (10.4)	9 (4.2)	17 (8.3)	52 (25.0)	47 (22.9)	35 (17.1)	25 (12.1)

Note. The correlation analysis of education with the age of the respondents in our study showed that primary education was marked by people aged up to 24 years of age.

**Table 3 ijerph-20-02245-t003:** Do you feel any fear of what your life will be like after the pandemic? (N = 1000).

	MD	25th–75th Percentiles	Kruskal–Wallis Test Value	SD	df	Mean (95% CI)	Mean Rank	χ^2^	*p* Value
**Sex**									
			2.901					4.292	0.637
Female	5.00	4.00–6.00		1.730	1	4.40 (4.25–4.55)	514.81		
Male	4.00	3.00–5.00		1.745		4.21 (4.05–4.37)	484.23		
**Age**									
			2.121					29.443	0.204
18–24	4.00	3.00–6.00		1.715	4	4.31 (4.00–4.61)	499.14		
25–34	5.00	4.00–5.00		1.649		4.49 (4.25–4.73)	524.23		
35–49	4.00	3.00–6.00		1.783		4.33 (4.10–4.56)	501.89		
50–64	4.00	3.09–5.00		1.712		4.35 (4.06–4.44)	490.27		
65 and more	4.00	3.00–5.66		1.866		4.17 (3.85–4.49)	486.32		
**Education**									
			3.678		3			14.066	0.725
Primary	4.00	3.00–5.00		1.548		4.05 (3.49–4.62)	442.58		
Vocational	5.00	4.00–6.00		1.685		4.56 (4.21–4.91)	539.57		
Secondary	4.00	3.00–5.00		1.780		4.26 (4.10–4.43)	491.55		
Tertiary	5.00	4.00–6.00		1.718		4.33 (4.17–4.50)	506.23		
**Place of residence**									
			4.596		5			34.609	0.257
Village (rural area)	5.00	4.00–5.00		1.662		4.35 (4.18–4.51)	500.16		
Town of 20 k citizens	5.00	4.00–6.00		1.705		4.56 (4.22–4.90)	541.85		
Town of 20–99 k citizens	5.00	4.00–6.00		1.828		4.33 (4.08–4.58)	507.18		
City of 100–199 k citizens	4.00	3.00–5.00		1.759		4.10 (3.72–4.48)	471.27		
City of 200–499 k citizens	5.00	4.00–6.00		1.772		4.32 (3.94–4.70)	506.93		
City of over 500 k citizens	4.00	3.00–5.00		1.820		4.11 (3.78–4.43)	471.86		
**Income**									
			5.214		5			22.314	0.842
>1000	5.00	4.00–6.00		1.709		4.69 (4.23–5.15)	561.68		
1000–2000	5.00	3.00–5.00		1.728		4.30 (4.07–4.54)	499.82		
2001–3000	5.00	3.00–5.00		1.878		4.19 (3.94–4.43)	489.10		
3001–5000	4.00	3.26–5.00		1.630		4.22 (3.99–4.45)	481.83		
<5000	4.00	3.31–5.00		1.656		4.31 (3.98–4.63)	487.19		
Refusal to answer	5.00	4.00–6.00		1.732		4.45 (4.22–4.69)	522.30		

Note: The values in the table are calculated based on an extensive, 7-point Likert scale, where 1 means “definitely not” and 7 means “definitely yes”.

**Table 4 ijerph-20-02245-t004:** Despondency: statistical analysis (N = 1000).

How Often Have You Felt Despondent Because of the Pandemic?								
	Mean (95% CI)	SD	Median	25th–75th Percentiles	df	Kruskal–Wallis Test Value	χ^2^	*p* Value
**Sex**								
					1	11.330	8.205	0.001
Female	4.56 (4.41–4.71)	1.764	5.00	3.00–6.00				
Male	4.19 (4.03–4.35)	1.760	4.00	3.00–5.22				
**Age**								
					4	12.483	3.827	0.014
18–24	4.80 (4.52–5.08)	1.592	5.00	4.00–6.00				
25–34	4.54 (4.30–4.78)	1.688	5.00	4.00–6.00				
35–49	4.41 (4.18–4.63)	1.784	5.00	3.00–6.00				
50–64	4.17 (3.97–4.38)	1.825	4.00	3.09–6.00				
65 and more	4.26 (3.95–4.57)	1.824	5.00	3.00–6.00				
**Education**								
					3	2.260	0.935	0.520
Primary	4.27 (3.67–4.87)	1.646	5.00	3.00–6.00				
Vocational	4.56 (4.16–4.96)	1.899	5.00	4.00–6.00				
Secondary	4.30 (4.14–4.46)	1.796	5.00	3.00–6.00				
Tertiary	4.46 (4.29–4.62)	1.722	5.00	3.00–6.00				
**Place of residence**								
					5	4.521	3.194	0.477
Village (rural area)	4.35 (4.17–4.52)	1.795	5.00	3.00–6.00				
Town of 20 k citizens	4.53 (4.21–4.86)	1.625	5.00	4.00–6.00				
Town of 20–99 k citizens	4.24 (3.98–4.50)	1.877	5.00	2.93–6.00				
City of 100–199 k citizens	4.29 (3.91–4.67)	1.772	5.00	3.00–6.00				
City of 200–499 k citizens	4.66 (4.29–5.03)	1.725	5.00	4.00–6.00				
City of over 500 k citizens	4.53 (4.23–4.82)	1.637	5.00	4.00–6.00				
**Income**								
					5	1.068	2.266	0.957
>1000	4.40 (3.93–4.87)	1.735	5.00	3.00–6.00				
1000–2000	4.41 (4.15–4.66)	1.869	5.00	3.00–6.00				
2001–3000	4.32 (4.09–4.55)	1.756	5.00	3.00–6.00				
3001–5000	4.47 (4.23–4.71)	1.698	5.00	4.00–6.00				
<5000	4.32 (3.99–4.66)	1.694	5.00	3.00–6.00				
Refusal to answer	4.39 (4.14–4.64)	1.813	5.00	3.00–6.00				

Note. The values in the table are calculated based on an extensive, 7-point Likert scale, where 1 meant “at all” and 7 meant “very often”.

**Table 5 ijerph-20-02245-t005:** Mental tension: statistical analysis (N = 1000).

How Often Have You Felt Psychologically under Pressure Due to the Pandemic?								
	Mean (95% CI)	SD	Median	25th–75th Percentiles	df	Kruskal–Wallis Test Value	χ^2^	*p* Value
**Sex**								
					1	6.448	1.483	0.011
Female	4.47 (4.32–4.62)	1.754	5.00	3.00–6.00				
Male	4.16 (3.99–4.33)	1.836	4.00	3.00–600				
**Age**								
					4	16.516	13.763	0.002
18–24	4.75 (4.45–5.05)	1.698	5.00	4.00–6.00				
25–34	4.42 (4.15–4.68)	1.840	5.00	3.00–6.00				
35–49	4.45 (4.22–4.68)	1.813	5.00	3.00–6.00				
50–64	4.09 (3.90–4.29)	1.742	4.00	3.00–5.00				
65 and more	4.12 (3.80–4.44)	1.854	4.00	3.00–6.00				
**Education**								
					3	2.937	2.237	0.401
Primary	4.23 (3.55–4.90)	1.859	4.15	3.00–6.00				
Vocational	4.37 (3.96–4.77)	1.942	5.00	3.00–6.00				
Secondary	4.22 (4.06–4.25)	1.797	4.00	3.00–6.00				
Tertiary	4.43 (4.27–4.60)	1.763	5.00	3.00–6.00				
**Place of residence**								
					5	6.296	5.373	0.278
Village (rural area)	4.31 (4.13–4.49)	1.833	4.00	3.00–6.00				
Town of 20 k citizens	4.54 (4.24–4.83)	1.473	5.00	4.00–6.00				
Town of 20–99 k citizens	4.13 (3.87–4.39)	1.885	4.00	3.00–5.74				
City of 100–199 k citizens	4.19 (3.80–4.58)	1.802	5.00	3.00–6.00				
City of 200–499 k citizens	4.59 (4.20–4.97)	1.809	5.00	4.00–6.00				
City of over 500 k citizens	4.43 (4.11–4.74)	1.758	5.00	3.00–6.00				
**Income**								
					5	4.495	6.154	0.481
>1000	4.76 (4.24–5.28)	1.915	5.00	4.00–6.00				
1000–2000	4.26 (4.01–4.51)	1.848	5.00	3.00–6.00				
2001–3000	4.27 (4.04–4.51)	1.778	5.00	3.00–6.00				
3001–5000	4.37 (4.13–4.62)	1.745	5.00	3.00–6.00				
<5000	4.29 (3.92–4.66)	1.852	4.00	3.00–6.00				
Refusal to answer	4.30 (4.06–4.54)	1.768	4.00	3.00–6.00				

Note. The values in the table are calculated based on an extensive, 7-point Likert scale, where 1 meant “at all” and 7 meant “very often”.

**Table 6 ijerph-20-02245-t006:** Fears for the future: statistical analysis (N = 1000).

How Often Have You Been Concerned about the Future Due to the Pandemic?								
	Mean (95% CI)	SD	Median	25th–75th Percentiles	df	Kruskal–Wallis Test Value	χ^2^	*p* Value
**Sex**								
					1	18.840	11.148	<0.001
Female	5.06 (4.92–5.20)	1.602	5.00	4.00–6.00				
Male	4.63 (4.48–4.78)	1.657	5.00	4.00–6.00				
**Age**								
					4	6.964	3.086	0.138
18–24	5.16 (4.89–5.43)	1.500	5.00	4.00–6.42				
25–34	4.92 (4.69–5.15)	1.609	5.00	4.00–6.00				
35–49	4.87 (4.66–5.08)	1.662	5.00	4.00–6.00				
50–64	4.70 (4.52–4.78)	1.672	5.00	4.00–6.00				
65 and more	4.84 (4.55–5.12)	1.678	5.00	4.00–6.00				
**Education**								
					3	5.264	7.427	0.153
Primary	5.12 (4.60–5.64)	1.440	5.00	4.00–6.00				
Vocational	5.06 (4.63–5.43)	1.796	5.08	4.00–6.25				
Secondary	4.78 (4.62–4.93)	1.659	5.00	4.00–6.00				
Tertiary	4.89 (4.73–5.04)	1.599	5.00	4.00–6.00				
**Place of residence**								
					5	2.995	2.910	0.701
Village (rural area)	4.81 (4.65–4.97)	1.627	5.00	4.00–6.00				
Town of 20 k citizens	5.00 (4.70–5.30)	1.519	5.00	4.00–6.00				
Town of 20–99 k citizens	4.79 (4.56–5.03)	1.716	5.00	4.00–6.00				
City of 100–199 k citizens	4.87 (4.49–5.25)	1.758	5.00	4.00–6.00				
City of 200–499 k citizens	5.04 (4.68–5.39)	1.650	5.00	4.00–6.00				
City of over 500 k citizens	4.89 (4.60–5.17)	1.581	5.00	4.00–6.00				
**Income**								
					5	15.651	12.458	0.008
>1000	5.44 (5.00–5.89)	1.622	6.00	5.00–7.00				
1000–2000	4.94 (4.72–5.17)	1.643	5.00	4.00–6.00				
2001–3000	4.80 (4.59–5.01)	1.646	5.00	4.00–6.00				
3001–5000	4.69 (4.47–4.91)	1.573	5.00	4.00–6.00				
<5000	4.66 (4.34–4.97)	1.589	5.00	4.00–6.00				
Refusal to answer	4.94 (4.71–5.18)	1.698	5.00	4.00–6.00				

Note. The values in the table are calculated based on an extensive, 7-point Likert scale, where 1 meant “at all” and 7 meant “very often”.

**Table 7 ijerph-20-02245-t007:** Insomnia: statistical analysis (N = 1000).

How Often Have You Had Problems Concentrating Due to the Insomnia?								
	Mean (95% CI)	SD	Median	25th–75th Percentiles	df	Kruskal–Wallis Test Value	χ^2^	*p* Value
**Sex**								
					1	4.713	2.004	0.030
Female	5.06 (4.92–5.20)	1.602	5.00	4.00–6.00				
Male	4.63 (4.48–4.78)	1.657	5.00	4.00–6.00				
**Age**								
					4	5.041	11.155	0.283
18–24	5.16 (4.89–5.43)	1.500	5.00	4.00–6.42				
25–34	4.92 (4.69–5.15)	1.609	5.00	4.00–6.00				
35–49	4.87 (4.66–5.08)	1.662	5.00	4.00–6.00				
50–64	4.70 (4.52–4.78)	1.672	5.00	4.00–6.00				
65 and more	4.84 (4.55–5.12)	1.678	5.00	4.00–6.00				
**Education**								
					3	4.471	3.264	0.215
Primary	5.12 (4.60–5.64)	1.440	5.00	4.00–6.00				
Vocational	5.06 (4.63–5.43)	1.796	5.08	4.00–6.25				
Secondary	4.78 (4.62–4.93)	1.659	5.00	4.00–6.00				
Tertiary	4.89 (4.73–5.04)	1.599	5.00	4.00–6.00				
**Place of residence**								
					5	3.517	5.606	0.621
Village (rural area)	4.81 (4.65–4.97)	1.627	5.00	4.00–6.00				
Town of 20 k citizens	5.00 (4.70–5.30)	1.519	5.00	4.00–6.00				
Town of 20–99 k citizens	4.79 (4.56–5.03)	1.716	5.00	4.00–6.00				
City of 100–199 k citizens	4.87 (4.49–5.25)	1.758	5.00	4.00–6.00				
City of 200–499 k citizens	5.04 (4.68–5.39)	1.650	5.00	4.00–6.00				
City of over 500 k citizens	4.89 (4.60–5.17)	1.581	5.00	4.00–6.00				
**Income**								
					5	6.403	3.804	0.269
>1000	5.44 (5.00–5.89)	1.622	6.00	5.00–7.00				
1000–2000	4.94 (4.72–5.17)	1.643	5.00	4.00–6.00				
2001–3000	4.80 (4.59–5.01)	1.646	5.00	4.00–6.00				
3001–5000	4.69 (4.47–4.91)	1.573	5.00	4.00–6.00				
<5000	4.66 (4.34–4.97)	1.589	5.00	4.00–6.00				
Refusal to answer	4.94 (4.71–5.18)	1.698	5.00	4.00–6.00				

Note: The values in the table are calculated based on an extensive, 7-point Likert scale, where 1 meant “at all” and 7 meant “very often”.

**Table 8 ijerph-20-02245-t008:** Concentration problems: statistical analysis (N = 1000).

How Often Have You Had Problems Concentrating Due to the Pandemic?								
	Mean (95% CI)	SD	Median	25th–75th Percentiles	df	Kruskal–Wallis Test Value	χ^2^	*p* Value
**Sex**								
					1	3.067	5.408	0.080
Female	4.08 (3.92–4.23)	1.811	4.00	3.00–5.00				
Male	3.91 (3.74–4.07)	1.820	4.00	2.00–5.00				
**Age**								
					4	25.445	19.163	<0.001
18–24	4.49 (4.19–5.79)	1.714	5.00	3.00–6.00				
25–34	4.24 (3.99–4.49)	1.764	4.00	3.00–6.00				
35–49	4.11 (3.88–4.33)	1.795	4.00	3.00–6.00				
50–64	3.68 (3.48–3.88)	1.816	4.00	2.00–5.00				
65 and more	3.74 (3.42–4.06)	1.871	4.00	2.00–5.00				
**Education**								
					3	4.489	4.169	0.180
Primary	4.30 (3.70–4.89)	1.633	4.00	3.91–5.41				
Vocational	4.09 (3.71–4.47)	1.826	4.00	3.00–5.00				
Secondary	3.87 (3.70–4.04)	1.824	4.00	2.00–5.00				
Tertiary	4.10 (3.92–4.27)	1.814	4.00	3.00–5.00				
**Place of residence**								
					5	7.907	4.637	0.161
Village (rural area)	3.99 (3.81–4.17)	1.841	4.00	2.00–5.00				
Town of 20 k citizens	4.11 (3.82–4.40)	1.467	4.00	3.00–5.00				
Town of 20–99 k citizens	3.77 (3.51–4.03)	1.716	4.00	2.00–5.00				
City of 100–199 k citizens	3.95 (3.54–4.36)	1.758	4.00	2.00–5.00				
City of 200–499 k citizens	4.38 (3.99–4.77)	1.824	5.00	3.37–6.00				
City of over 500 k citizens	4.08 (3.76–4.41)	1.785	4.00	3.00–5.00				
**Income**								
					5	3.886	1.939	0.566
>1000	3.94 (3.42–4.46)	1.917	4.00	2.00–6.00				
1000–2000	4.00 (3.74–4.26)	1.879	4.00	2.00–5.00				
2001–3000	3.81 (3.57–4.05)	1.819	4.00	2.00–5.00				
3001–5000	4.07 (3.83–4.31)	1.718	4.00	3.00–5.00				
<5000	4.18 (3.82–4.53)	1.803	4.00	3.00–6.00				
Refusal to answer	4.07 (3.82–4.32)	1.822	4.00	2.00–6.00				

Note: The values in the table are calculated based on an extensive, 7-point Likert scale, where 1 meant “at all” and 7 meant “very often”.

**Table 9 ijerph-20-02245-t009:** Depressive mood—statistical analysis (N = 1000).

How Often Have You Felt Depressed Because of the Pandemic?								
	Mean (95% CI)	SD	Median	25th–75th Percentiles	df	Kruskal–Wallis Test Value	χ^2^	*p* Value
**Sex**								
					1	11.330	2.662	0.050
Female	4.09 (3.93–4.25)	1.896	4.00	2.00–6.00				
Male	3.87 (3.69–4.04)	1.930	4.00	2.00–5.00				
**Age**								
					4	13.364	6.635	0.010
18–24	4.40 (4.10–4.70)	1.709	4.00	3.00–6.00				
25–34	4.13 (3.86–4.39)	1.867	4.00	3.00–5.00				
35–49	4.11 (3.87–4.35)	1.929	4.00	2.00–6.00				
50–64	3.73 (3.51–3.95)	1.932	4.00	2.00–5.00				
65 and more	3.77 (3.43–4.11)	2.009	4.00	2.00–6.00				
**Education**								
					3	2.075	1.565	0.557
Primary	3.98 (3.33–4.63)	1.784	4.00	3.00–5.00				
Vocational	4.11 (3.70–4.51)	1.949	4.00	2.00–6.00				
Secondary	3.89 (3.72–4.07)	1.926	4.00	2.00–5.00				
Tertiary	4.06 (3.88–4.25)	1.905	4.00	3.00–6.00				
**Place of residence**								
					5	10.454	7.065	0.063
Village (rural area)	3.98 (3.79–4.17)	1.927	4.00	2.00–5.00				
Town of 20 k citizens	4.23 (3.87–4.59)	1.820	4.00	3.00–6.00				
Town of 20–99 k citizens	3.82 (3.55–4.09)	1.950	4.00	2.00–5.00				
City of 100–199 k citizens	3.60 (3.18–4.03)	1.975	4.00	1.99–5.00				
City of 200–499 k citizens	4.36 (3.98–4.75)	1.810	5.00	3.00–6.00				
City of over 500 k citizens	4.10 (3.76–4.43)	1.872	4.00	3.00–5.35				
**Income**								
					5	3.397	2.802	0.639
>1000	4.12 (3.58–4.66)	2.003	4.30	2.00–6.00				
1000–2000	4.08 (3.81–4.36)	2.007	4.00	2.00–6.00				
2001–3000	3.80 (3.55–4.06)	1.950	4.00	2.00–5.00				
3001–5000	4.03 (3.77–4.29)	1.844	4.00	3.00–5.00				
<5000	4.07 (3.71–4.43)	1.807	4.00	3.00–5.34				
Refusal to answer	3.97 (3.71–4.23)	1.880	4.00	2.00–5.00				

Note: The values in the table are calculated based on an extensive, 7-point Likert scale, where 1 meant “at all” and 7 meant “very often”.

**Table 10 ijerph-20-02245-t010:** Breathing problems: statistical analysis (N = 1000).

How Often Have You Had Breathing Problems Due to the Pandemic?								
	Mean (95% CI)	SD	Median	25th–75th Percentiles	df	Kruskal–Wallis Test Value	χ^2^	*p* Value
**Sex**								
					1	0.232	0.674	0.630
Female	3.09 (2.92–3.25)	1.925	3.00	1.00–5.00				
Male	3.17 (2.99–3.34)	1.949	3.00	1.00–5.00				
**Age**								
					4	7.527	6.498	0.111
18–24	3.18 (2.86–3.51)	1.826	3.00	1.00–5.00				
25–34	3.18 (2.89–3.47)	2.016	3.00	1.00–5.00				
35–49	3.37 (3.12–3.62)	1.994	3.00	1.00–5.00				
50–64	2.91 (2.70–3.11)	1.855	3.00	1.00–4.00				
65 and more	3.06 (2.72–3.39)	1.959	2.77	1.00–4.00				
**Education**								
					3	3.628	3.036	0.305
Primary	3.54 (2.88–4.19)	1.803	3.00	2.00–5.49				
Vocational	3.34 (2.94–3.74)	1.913	3.96	1.00–5.00				
Secondary	3.05 (2.87–3.23)	1.915	3.00	1.00–4.00				
Tertiary	3.13 (2.94–3.32)	1.972	3.00	1.00–5.00				
**Place of residence**								
					5	8.270	9.793	0.142
Village (rural area)	3.07 (2.88–3.27)	1.965	3.00	3.00–5.00				
Town of 20 k citizens	3.29 (2.92–365)	1.851	3.00	3.00–5.00				
Town of 20–99 k citizens	2.98 (2.72–3.24)	1.889	3.00	3.00–4.00				
City of 100–199 k citizens	2.87 (2.47–3.27)	1.975	3.00	3.00–4.00				
City of 200–499 k citizens	3.49 (3.09–3.90)	1.901	4.00	4.00–5.00				
City of over 500 k citizens	3.33 (2.96–3.69)	2.039	3.35	3.35–5.00				
**Income**								
					5	8.703	11.329	0.121
>1000	2.90 (2.32–3.48)	2.120	2.00	1.00–4.44				
1000–2000	3.06 (2.81–3.32)	1.880	3.00	1.00–4.00				
2001–3000	3.00 (2.75–3.26)	1.948	3.00	1.00–5.00				
3001–5000	3.17 (2.91–3.43)	1.870	3.00	1.00–5.00				
<5000	3.64 (3.23–4.05)	2.080	4.00	1.00–5.00				
Refusal to answer	3.09 (2.83–3.35)	1.897	3.00	1.00–4.00				

Note: The values in the table are calculated based on an extensive, 7-point Likert scale, where 1 meant “at all” and 7 meant “very often”.

**Table 11 ijerph-20-02245-t011:** Chest pain: statistical analysis (N = 1000).

How Often Did You Experience Chest Pains Due to the Pandemic?								
	Mean (95% CI)	SD	Median	25th–75th Percentiles	df	Kruskal–Wallis Test Value	χ^2^	*p* Value
**Sex**								
					1	1.309	3.044	0.253
Female	3.18 (3.02–3.35)	1.931	3.00	1.00–5.00				
Male	3.32 (3.15–3.50)	1.933	3.00	1.00–5.00				
**Age**								
					4	4.607	4.386	0.330
18–24	3.41 (3.07–3.74)	1.890	3.00	1.00–5.00				
25–34	3.35 (3.06–3.63)	1.998	3.62	1.00–5.00				
35–49	3.36 (3.11–3.60)	1.962	4.00	1.00–5.00				
50–64	3.14 (2.93–3.35)	1.886	3.00	1.00–5.00				
65 and more	3.02 (2.69–3.34)	1.918	2.75	1.00–4.75				
**Education**								
					3	7.892	4.973	0.048
Primary	3.99 (3.28–4.71)	1.969	4.00	2.20–6.00				
Vocational	3.54 (3.13–3.95)	1.955	4.00	2.00–5.00				
Secondary	3.15 (2.98–3.33)	1.918	3.00	1.00–5.00				
Tertiary	3.24 (3.05–3.42)	1.929	3.00	1.00–5.00				
**Place of residence**								
					5	7.563	7.236	0.182
Village (rural area)	3.26 (3.07–3.45)	1.920	3.00	1.00–5.00				
Town of 20 k citizens	3.41 (3.04–3.77)	1.844	4.00	1.97–5.00				
Town of 20–99 k citizens	3.07 (2.81–3.33)	1.918	3.00	1.00–5.00				
City of 100–199 k citizens	2.93 (2.52–3.34)	1.891	2.79	1.00–4.22				
City of 200–499 k citizens	3.38 (2.95–3.80)	1.996	4.00	1.00–5.00				
City of over 500 k citizens	3.51 (3.15–3.88)	2.082	4.00	1.00–5.00				
**Income**								
					5	9.345	10.325	0.096
>1000	3.24 (2.68–3.50)	2.060	3.00	1.00–5.00				
1000–2000	3.07 (2.81–3.33)	1.907	3.00	1.00–5.00				
2001–3000	3.07 (2.82–3.31)	1.895	3.00	1.00–5.00				
3001–5000	3.33 (3.06–3.59)	1.890	4.00	1.00–5.00				
<5000	3.65 (3.26–4.04)	1.987	4.00	1.00–5.00				
Refusal to answer	3.37 (3.10–3.63)	1.956	3.00	1.00–5.00				

Note. The values in the table are calculated based on an extensive, 7-point Likert scale, where 1 meant “at all” and 7 meant “very often”.

**Table 12 ijerph-20-02245-t012:** Digestive problems: statistical analysis (N = 1000).

How Often Have You Had Digestive Problems Due to the Pandemic?								
	Mean (95% CI)	SD	Median	25th–75th Percentiles	df	Kruskal–Wallis Test Value	χ^2^	*p* Value
**Sex**								
					1	0.673	0.297	0.412
Female	3.56 (3.40–3.72)	1.865	4.00	2.00–5.00				
Male	3.47 (3.30–3.64)	1.841	4.00	2.00–5.00				
**Age**								
					4	5.037	7.674	0.284
18–24	3.62 (3.30–3.94)	1.791	4.00	2.00–5.00				
25–34	3.61 (3.33–3.88)	1.923	4.00	2.00–5.00				
35–49	3.66 (3.41–3.90)	1.922	4.00	2.00–5.00				
50–64	3.34 (3.14–3.53)	1.777	3.00	2.00–5.00				
65 and more	3.45 (3.13–3.76)	1.846	3.00	2.00–5.00				
**Education**								
					3	1.496	1.497	0.683
Primary	3.65 (3.01–4.28)	1.741	4.00	2.00–5.00				
Vocational	3.72 (3.34–4.10)	1.810	4.00	2.74–5.00				
Secondary	3.48 (3.31–3.65)	1.837	4.00	2.00–5.00				
Tertiary	3.50 (3.32–3.68)	1.891	4.00	2.00–5.00				
**Place of residence**								
					5	5.672	2.762	0.339
Village (rural area)	3.50 (3.32–3.69)	1.856	4.00	2.00–5.00				
Town of 20 k citizens	3.69 (3.38–4.01)	1.580	4.00	2.07–5.00				
Town of 20–99 k citizens	3.35 (3.09–3.61)	1.861	3.00	2.00–5.00				
City of 100–199 k citizens	3.41 (3.00–3.81)	1.884	4.00	1.00–5.00				
City of 200–499 k citizens	3.87 (3.45–4.29)	1.966	5.00	2.00–5.24				
City of over 500 k citizens	3.51 (3.16–3.85)	1.929	4.00	2.00–5.00				
**Income**								
					5	4.691	3.739	0.445
>1000	3.24 (2.72–3.76)	1.921	3.00	1.00–5.00				
1000–2000	3.44 (3.20–3.69)	1.782	4.00	2.00–5.00				
2001–3000	3.40 (3.15–3.65)	1.921	3.00	1.00–5.00				
3001–5000	3.61 (3.35–3.87)	1.838	4.00	2.00–5.00				
<5000	3.70 (3.34–4.06)	1.805	4.00	2.00–5.00				
Refusal to answer	3.61 (3.35–3.87)	1.869	4.00	2.00–5.00				

Note. The values in the table are calculated based on an extensive, 7-point Likert scale, where 1 meant “at all” and 7 meant “very often”.

## Data Availability

The data presented in this study are available on request from the corresponding author.

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
