# Peer review of "Shades of Fear—Mental and Physical Health Reactions to the COVID-19 Pandemic: A Representative Study of Polish Society"

_ijerph, 2023, doi:10.3390/ijerph20032245_

Round 1

Reviewer 1 Report

The study examined the state of psychological and physiological of the fear on COVID-19 in Poland. Detailed analysis showed that responses in a pandemic are strongly determined by demographic categories, mainly 23 gender, and age, and differ depending on the social group to which a person belongs. There must be some hypothesis those explain the differences in the degree of fear on Pandemic. For example, those who don’t have the correct information tends to have some psychological tension from the fear, but those who could analyze the risk of the virus from the scientific data has less probability of feeling fear on COVID-19. To examine the hypothesis, some questions on the media of getting information and the accessibility to the correct information would be needed.  

  In addition, the process of COVID-19 pandemic has a quite complicated, and the responses of people has changed drastically according to the degree of risks of the virus. The risks of virus have decreased as time passed because of the variants of the virus prevailed. In addition, it has been proved that the vaccination had negative effects on the prevention of virus from several reasons such as ADE (anti-body dependent enhancement) and the variants of virus. For example, the ratio of booster vaccination in Japan is the highest in the world, and the ration of cases confirmed per 100 thousand people became the highest in the world in 2022.

In addition, it is proved that Swedish strategy which adopted the less rigid prevention measure resulted in the better social situation with less costs, and lockdown measure had less effective for the prevention and brought about high cost.

 I argue that the state of fear depends not only demographic factors such as age or sex, but also on public policy or social factors such as provision of information. In my understanding, the discussions about the issues written in the above should be included so that the paper has a social values and effective policy implications.

Author Response

Response to Reviewer 1.

Thank you very much for carefully reading our manuscript. We also thank you for the insights you provided. We have referred to them below. Thanks to them, we have also made significant corrections to our article.

The study examined the state of psychological and physiological of the fear on COVID-19 in Poland. Detailed analysis showed that responses in a pandemic are strongly determined by demographic categories, mainly 23 gender, and age, and differ depending on the social group to which a person belongs. There must be some hypothesis those explain the differences in the degree of fear on Pandemic. For example, those who don’t have the correct information tends to have some psychological tension from the fear, but those who could analyze the risk of the virus from the scientific data has less probability of feeling fear on COVID-19. To examine the hypothesis, some questions on the media of getting information and the accessibility to the correct information would be needed.  

Thank you very much for this insight. Indeed, recently, after the end of the pandemic, studies have begun to appear that indicate that the level of anxiety is related to the consumption of media, especially social media, such as Facebook, Instagram TikTok, in which there was a lot of disinformation. It is impacted the higher level of anxiety in the audience of these media. In traditional media, such as radio and television, information is verified, and disinformation appears much less frequently. Perhaps this is some explanation for why in our study, younger people were statistically significantly more likely to show higher pandemic-related anxiety than older people (more likely to use traditional media). Nonetheless, we did not study this, and therefore cannot state this for sure. We have added the Limitations of this thesis with an indication for further research. Lines:

In addition, the process of COVID-19 pandemic has a quite complicated, and the responses of people has changed drastically according to the degree of risks of the virus. The risks of virus have decreased as time passed because of the variants of the virus prevailed. In addition, it has been proved that the vaccination had negative effects on the prevention of virus from several reasons such as ADE (anti-body dependent enhancement) and the variants of virus. For example, the ratio of booster vaccination in Japan is the highest in the world, and the ration of cases confirmed per 100 thousand people became the highest in the world in 2022. 

In our study, we focused on the first year of the pandemic, even before the introduction of widespread vaccination; only selected groups of people were vaccinated at the time of the study, as mentioned in the manuscript. The invention of the vaccine may have been a triggering factor in lowering the sense of fear of the pandemic. At the same time, it may have been joined by fear of vaccines, but we did not ask about fear of vaccines in our study, as this was not the focus of our investigation. Nevertheless, this is a very interesting topic for research and analysis. 

In addition, it is proved that Swedish strategy which adopted the less rigid prevention measure resulted in the better social situation with less costs, and lockdown measure had less effective for the prevention and brought about high cost. 

The use of various methods of prevention from infections is being studied in other papers especially after the pandemic. When we implemented our study, it took a lot of work to assess and capture. Nonetheless, this comment helped us a lot because we realized that it would be helpful to include in our article a short calendar of the strictures implemented in Poland, which could help in the perception of the results of our research. Lines:

I argue that the state of fear depends not only demographic factors such as age or sex, but also on public policy or social factors such as provision of information. In my understanding, the discussions about the issues written in the above should be included so that the paper has a social values and effective policy implications.

We fully agree with this opinion. Many factors influence a sense of insecurity, and our study aimed to learn how demographic factors affect this. In the research we plan to conduct this year, three years after the pandemic announcement, we will undoubtedly take this cue and deepen it with political and social factors using the knowledge from this research.

Thank you once again!

Reviewer 2 Report

Dear Editors and authors,

Thank you for inviting me to review this manuscript, and congratulations for your work. This is a relevant exploration to understand how the COVID-19 Pandemic has affected mental and physical state of Polish population. However, there are some opportunities for improvement, as is outlined below.

General

Headlines of each section are numbered as "1", please number each section correctly

Abstract

Line 12: Please explain the abbreviations “EU” and “CAWI”, as their first appearance is in the abstract.

Keywords:

Line 25: "Fear" may not be the best keyword, since it already appears in the title, so the authors are losing the opportunity to choose another word that facilitates reading and citing their work.

Introduction

Lines 33-34: Are not written in English, please translate them.

Line 38: “that are different from each other because of demographics” “that have differences due to demographics”.

Line 46: Instead of “were studied”, it could be preferrable to say “have being studied”.

Lines 64-65: Is the following statement true “and a still small number of studies on the impact of COVID-19 pandemics on mental and physical health societies”? Verify it or remove it.

Lines 77-78: Please respect the format of the journal, regarding the indentation.

Further context could be given in the introduction, the authors could address the impact of the lockdown on the physical activity levels of the population, since it affects both physical and psychological state, and a large body of evidence is available.

Methods

Population

Lines 81-101: These two paragraphs should be in the "Introduction section". And references that corroborate the presented date need to be provided.

The population included in the study is not described, at least it should be said that further information is provided in Table 1.

Line 108: I suggest that the section "Implementation of the study and sample size" is removed, and that the paragraph included there is moved to the "Population section"

Line 108: This data should be presented as "n=1.000"

Line 156: Explain the abbreviation “SW Research”.

Results

Table 1: What were the criteria used to choose the age limits, to configure the age groups?

Table 2: It should be stated that data is presented in the following format "N(%)"

Figure 1: The authors should corect "Menatal tension" and write "Mental tension"

Discussion

Lines 262-263: The statement made in this sentence may be too strong and may not be accurate considering the present results.

Line 284: It is relevant to indicate the main author of this article?. If it is, is stated in the correct way?.

Line 297: It may be advisable to remove the word “thesis”.

Lines 296-303: Please compare your results with other research that address anxiety disorders during the pandemic period.

Line 330: Instead of “women are a group that feels stronger fear of the pandemic than men” it may be better to say: “women group feels stronger fear of the pandemic than men.”

Line 350: Maybe a more indirect writing style could be better, for instance “The present study” instead of “Our study”.

Line 361: A sample size calculation should be conducted to demonstrate that the study is in fact representative.

Line 382: The sentence “in the future pandemic crisis” could be replaced for “in future pandemic crisis”.

Author Response

Response to Reviewer 2.

Thank you very much for carefully reading our manuscript. We also thank you for the insights you provided. We have referred to them below. Thanks to them, we have also made significant corrections to our article.

Thank you for inviting me to review this manuscript, and congratulations for your work. This is a relevant exploration to understand how the COVID-19 Pandemic has affected mental and physical state of Polish population. However, there are some opportunities for improvement, as is outlined below.

General

Headlines of each section are numbered as "1", please number each section correctly

Thank you. We have corrected it throughout the whole manuscript.

Abstract

Line 12: Please explain the abbreviations “EU” and “CAWI”, as their first appearance is in the abstract.

Thank you. We have expanded the abbreviations. Lines: 13 and 14

Keywords:

Line 25: "Fear" may not be the best keyword, since it already appears in the title, so the authors are losing the opportunity to choose another word that facilitates reading and citing their work.

It is a very good point. Thank you for that. We have changed it. Line: 27

Introduction

Lines 33-34: Are not written in English, please translate them.

We are sorry for that. We have changed it. Lines: 35-37

Line 38: “that are different from each other because of demographics” “that have differences due to demographics”.

Thank you. We have corrected it. Line: 40

Line 46: Instead of “were studied”, it could be preferrable to say “have being studied”.

Thank you. We have corrected it. Line: 58

Lines 64-65: Is the following statement true “and a still small number of studies on the impact of COVID-19 pandemics on mental and physical health societies”? Verify it or remove it.

Thank you. We have decided to remove that part of the sentence.

Lines 77-78: Please respect the format of the journal, regarding the indentation.

We corrected it.

Further context could be given in the introduction, the authors could address the impact of the lockdown on the physical activity levels of the population, since it affects both physical and psychological state, and a large body of evidence is available.

We realize that the Introduction could have contained even more information and references to the literature on the subject. However, we felt that the article was already very long, and therefore decided not to expand this thread. Especially since throughout the paper, we refer to nearly sixty papers and studies from around the world that refer to this topic. 

Methods

Population

Lines 81-101: These two paragraphs should be in the "Introduction section". And references that corroborate the presented date need to be provided.

Thank you very much for this indication. As we have added a short calendar of the COVID-19 pandemic-related restrictions to the manuscript, which may provide a good socio-political background for the research results discussed, we decided to include one more chapter in the article: First year of the COVID-19 pandemic in Poland. Line: 90, 124-140

The population included in the study is not described, at least it should be said that further information is provided in Table 1.

We have added a reference to Table 1, thank you. Lines: 154-155

Line 108: I suggest that the section "Implementation of the study and sample size" is removed, and that the paragraph included there is moved to the "Population section"

Thank you very much for this indication. For more clarity on the content of this paragraph, we have decided to leave it here in the manuscript but have renamed it to Design of the Study and Sample Size. Line: 142

Line 108: This data should be presented as "n=1.000"

We have changed it. Line: 148

Line 156: Explain the abbreviation “SW Research”.

This is the official, proprietary name of the research company. Perhaps it is an acronym, of the type: HBO Max. 

Results

Table 1: What were the criteria used to choose the age limits, to configure the age groups?

In selecting the age categories, we considered the social and cultural references that apply in Poland and the research assumptions derived from the hypotheses. The first group-18-24-are schoolchildren, students, and young people starting their careers. The 25-34 group has completed their formal education, and the upper limit of 34 is the designation of the period of youth formally recognized in Poland. The 35-49 group comprises middle-aged people who usually seek personal AND professional stability. The 50-64 group are mature people reaching their professional and personal peaks. The last group is retirees, the age at which retirement is most common in Poland.

Table 2: It should be stated that data is presented in the following format "N(%)"

Thank you. We have added a corresponding line in Table 2 Line: 234

Figure 1: The authors should corect "Menatal tension" and write "Mental tension"

Thank you. We have changed it. Line: 270

Discussion

Lines 262-263: The statement made in this sentence may be too strong and may not be accurate considering the present results.

We have softened the overtones of this sentence. Thank you. Lines: 403-404

Line 284: It is relevant to indicate the main author of this article?. If it is, is stated in the correct way?.

Thank you. We have changed it. Lines: 425-427

Line 297: It may be advisable to remove the word “thesis”.

Thank you. We have removed the word. Line: 455

Lines 296-303: Please compare your results with other research that address anxiety disorders during the pandemic period.

We have added references to papers that address adaptation to stressful conditions. Unfortunately, we did not find papers that referenced representative studies in other societies one year after the pandemic outbreak that we could discuss this part with. Line: 460

Line 330: Instead of “women are a group that feels stronger fear of the pandemic than men” it may be better to say: “women group feels stronger fear of the pandemic than men.”

Thank you for this tip. We have changed this sentence. Line: 489

Line 350: Maybe a more indirect writing style could be better, for instance “The present study” instead of “Our study”.

We have changed it. Line: 557

Line 361: A sample size calculation should be conducted to demonstrate that the study is in fact representative.

The survey is representative of the Polish society, and the sample, as we wrote about in the Methods was selected in a random-quota fashion using a panel of the online research form SW Research, which conducted the survey.

Line 382: The sentence “in the future pandemic crisis” could be replaced for “in future pandemic crisis”.

Thank you. We have changed it. Line: 598

Thank you very much again for all pointers and hints.

Round 2

Reviewer 1 Report

The paper is revised well.